# CRISPR/Cas9 Mediated Knock Down of δ-ENaC Blunted the TNF-Induced Activation of ENaC in A549 Cells

**DOI:** 10.3390/ijms22041858

**Published:** 2021-02-12

**Authors:** Waheed Shabbir, Nermina Topcagic, Mohammed Aufy, Murat Oz

**Affiliations:** 1Department of Pharmacology and Toxicology, University of Vienna, Althanstrasse 14, A-1090 Vienna, Austria; nermina.topcagic@univie.ac.at (N.T.); mohammed.aufy@univie.ac.at (M.A.); 2Division of Nephrology and Cellular and Molecular Pharmacology, Department of Medicine, University of California, San Francisco, CA 94158, USA; 3Department of Pharmacology and Therapeutics, Faculty of Pharmacy, Kuwait University, Safat, Kuwait City 13110, Kuwait; ahmet.oz@ku.edu.kw

**Keywords:** tumor necrosis factor (TNF), epithelial sodium channel (ENaC), CRISPR/Cas9

## Abstract

Tumor necrosis factor (TNF) is known to activate the epithelial Na^+^ channel (ENaC) in A549 cells. A549 cells are widely used model for ENaC research. The role of δ-ENaC subunit in TNF-induced activation has not been studied. In this study we hypothesized that δ-ENaC plays a major role in TNF-induced activation of ENaC channel in A549 cells which are widely used model for ENaC research. We used CRISPR/Cas 9 approach to knock down (KD) the δ-ENaC in A549 cells. Western blot and immunofluorescence assays were performed to analyze efficacy of δ-ENaC protein KD. Whole-cell patch clamp technique was used to analyze the TNF-induced activation of ENaC. Overexpression of wild type δ-ENaC in the δ-ENaC KD of A549 cells restored the TNF-induced activation of whole-cell Na^+^ current. Neither N-linked glycosylation sites nor carboxyl terminus domain of δ-ENaC was necessary for the TNF-induced activation of whole-cell Na^+^ current in δ-ENaC KD of A549 cells. Our data demonstrated that in A549 cells the δ-ENaC plays a major role in TNF-induced activation of ENaC.

## 1. Introduction

To date, four epithelial sodium channel (ENaC α, β, γ and δ subunits) have been cloned in mammals [1,2,3]. The widely accepted concept is that the functional ENaCs must be composed of at least one pore forming α or α-like subunit and δ or δ-like subunits [4]. δ-ENaC expression has been found in sevral human tissues including gonads, pancreas, brain, heart, liver and thymus, with lower amounts in kidney and lung [3,4]. The function of δ-ENaC subunits in these tissues are not well studied. Some studies have shown that δ-ENaC containing ENaC have very slow activation and desensitization kinetics in response to decrease in extracellular pH [5] supporting their role as slow pH sensors which may be found in ischemia [4]. δ-ENaC channels was shown to contribute ~50% of amiloride-sensitive salt transport across primary human nasal epithelial cells [6].

The best possible characterization of the ENaC similar to native Alveolar Type II (ATII) cells was always a significant challenge. These studies have been obstructed by the fact that ATII cells are difficult to isolate and maintain in primary culture [7]. Additionally, these cells has been shown to spontaneously change their phenotype to Alvelolar type I cells [8]. These complications led the discovery of A549 cells. The A549 cells that originated from a human alveolar cell adeno carcinoma and possess many characteristics of native type II cells including multilamellar cytoplasmic inclusion bodies and the ability to synthesize surfactant phospholipids [9]. These cells express a 8.6-pS amiloride-sensitive Na^+^ channel with biophysical properties similar to those found in ATII cells in primary culture [7]. These characteristics made A459 best suitable model for ENaC research.

It has been shown that all α, β, γ and δ subunits of ENaC are expressed in A549 cells. Proinflammatory cytokine tumor necrosis factor (TNF) plays a dichotomous role in the reabsorption of pulmonary edema [10]. TNF has been shown to inhibit the mRNA of all ENaC subunits [11]. On the other hand, TNF has also known to activate whole-cell Na^+^ current through ENaC complex in A549 cells [12,13]. In earlier studies, TNF has been shown to exert its inhibitory effect on ENaC mRNA transcription by binding to its cognate receptors [11] and stimulatory effect through its lectin-like domain which is distinct from the receptor binding domain [14,15].

TNF lectin-like domain mimicking peptide (Solnatide) exerts its ENaC activating effect predominantly through either α, and/or δ-ENaC subunit in overexpression systems [13,16] and A549 cells endogenously expressing ENaC [12,13]. Different groups have shown that δ-ENaC overexpression stimulates the whole-cell ENaC current up to 11 fold compared to ENaC currents through α, β, γ subunit complex in a heterologous expression system [17]. These findings indicate that ENaC δ subunit plays a stimulatory role in the activation of ENaC channel complex. Activation of ENaC current by TNF in A549 cells has been widely studied [12,13], however the role of δ subunit in the activation of ENaC channel complex in A549 cells remains unknown.

This study was undertaken to analyze whether knock down (KD) of δ-ENaC can affect the TNF-induced activation of Na^+^ currents in A549 cells. CRISPR/Cas 9 approach was used to KD of δ-ENaC in A549 cells. Western blot and immunofluorescence assays were performed to analyze efficacy of δ-ENaC protein KD. Patch clamp assays were performed to analyze the ENaC channel function. TNF-induced activation of whole-cell Na^+^ current was inhibited in δ-ENaC KD A549 cells. Overexpression of wild type δ-ENaC in the δ-ENaC KD of A549 cells restored the TNF-induced activation of whole-cell Na^+^ current. We further analyzed the role of N-linked glycosylation sites and carboxyl terminus domain of δ-ENaC in TNF-induced activation of whole-cell Na^+^ current in δ-ENaC KD of A549 cells. Neither N-linked glycosylation sites nor carboxyl terminus domain was required to restore the TNF-induced activation of whole cell Na^+^ current in δ-ENaC KD of A549 cells. Our results provide new insights into the TNF-induced activation of ENaC in endogenously expressed A549 cells.

## 2. Results

### 2.1. CRISPR/Cas 9 Significantly Knocked Down the δ-ENaC Subunit Protein in the A549 and Cells

ENaC pore forming δ subunit is expressed in the A549 cells [4] and it has been shown that when co-expressed with βγ-ENaC can increase the macroscopic current two to 11 fold higher than αβγ in overexpression system [17]. To analyze the role of δ-ENaC in TNF-mediated activation of ENaC, we knocked down the δ-ENaC with CRISPR/Cas 9 plasmids in A549 cells. The δ-ENaC CRISPR/Cas 9 KO plasmids decreased 90% of the δ-ENaC protein expression in A549 cell compared with control CRISPR/Cas 9 plasmid, as shown in western blot and immunofluorescence assays (Figure 1A,B). These results indicate that δ-ENaC CRISPR/Cas 9 KO plasmid significantly knocked down the δ-ENaC subunit in A549 cells.

### 2.2. Knock Down of δ-ENaC Abolished the TNF-Mediated Activation of ENaC in A549 Cells

It has been previously shown that TNF can activate the ENaC current in A549 cells [12]. We have previously shown that TNF lectin-like domain mimicking peptide required either the pore-forming α-or δ-ENaC subunits for its current activating effect on ENaC when expressed in HEK-293 cells [13]. In this study we analyzed the δ-subunit mediated ENaC current activating effect of TNF through CRISPR/Cas9 knock down (KD) approach. Knock down of δ-ENaC caused significant decrease of control ENaC current in A549 cells assayed by whole-cell patch clamp configuration (Figure 2A,B). Additionally, 10 nM TNF was unable to activate the whole-cell ENaC current in these cells (Figure 2A,B). Whereas, the cells transfected with control CRISPR/Cas9 plasmids showed an activation of ENaC current with 10 nM TNF (Figure 2A,B). These results indicate that δ-ENaC plays a significant role in TNF-mediated activation of ENaC in A549 cells.

### 2.3. Overexpression of WT δ-ENaC Restored the TNF-Mediate Activation of ENaC Current in δ-ENaC Knock Down A549 Cells

To analyze whether the WT δ-ENaC overexpression restore the TNF-mediated activation of ENaC in δ-ENaC KD A549, we transiently transfected WT δ-ENaC in the A549 cells 24 h following the CRISPR/Cas9 knock down of δ-ENaC. Overexpression of WT δ-ENaC restored the 10 nM TNF-mediated activation of whole-cell current of ENaC (Figure 3A,B). These results indicate that a full length δ-ENaC is required for the TNF-mediated activation of whole-cell current in the δ-ENaC KD A549 cells.

### 2.4. N-Linked Glycosylation of δ-ENaC Is Not Necessary to Restore the TNF-Mediate Activation of ENaC Current in δ-ENaC Knock Down A549 Cells

We have previously shown that TNF lectin-like domain mimicking peptide required N-linked glycosylation sites of δ-ENaC to activate the ENaC channel in overexpression system [18]. Additionally, we have also shown that in A549 cells TNF can positively increase single channel kinetics without affecting conductivity of the channel and this increase was completely abolished in PNGase F pretreated cells [13]. Glycosylation sites on the ENaC subunits are necessary for TNF-induced activation of ENaC current and membrane insertion [19].

To analyze whether N-linked glycosylation sites of δ-ENaC are necessary to restore the ENaC channel activation by TNF in δ-ENaC KD A549 cells, we transiently transfected the A549 cells with δ-ENaC in which all three N-linked glycosylation sites (N166, 211, 384Q) were mutated (3NQ) 24 h following the CRISPR/Cas9 KD δ-ENaC. Overexpression of 3NQ δ-ENaC restored the 10 nM TNF-mediated activation of whole-cell ENaC currents in δ-ENaC KD A549 cells (Figure 4A,B). These results indicate N-Linked glycosylation of δ-ENaC is not necessary to restore the TNF-mediate activation of ENaC current in δ-ENaC KD A549 cells.

### 2.5. Both N-Linked Glycosylation and Carboxyl Terminus of δ-ENaC Is Not Necessary to Restore the TNF-Mediate Activation of ENaC Current in δ-ENaC Knock Down A549 Cells

We have previously shown that TNF lectin-like domain mimicking peptide required N-linked glycosylation sites as well as carboxyl terminus of δ-ENaC to activate the ENaC channel [18]. To analyze the role of N-linked glycosylation and carboxyl terminus of δ-ENaC in restoring the TNF-mediated activation of ENaC current in δ-ENaC KD A549 cells, we transiently transfected the A549 cells with δ3NQ-D522X-ENaC (δ-ENaC in which all N-linked glycosylation were mutated and carboxyl terminus was deleted). Twenty-four hours following the CRISPR/Cas9 KD δ-ENaC, overexpression of δ3NQ-D522X-ENaC restored the TNF-mediated activation of whole-cell current of ENaC (Figure 5A,B). These results indicate that both N-linked glycosylation and carboxyl terminus of δ-ENaC is not necessary to restore the TNF-mediated activation of ENaC current in δ-ENaC KD A549 cells.

## 3. Discussion

The aim of the present study was to test the hypothesis that the δ-ENaC subunit may contribute as the major subunit mediating the TNF-induced activation of ENaC channel complex in A549 cells.

A549 cells are commonly used in ENaC research. They have been used to analyze the acute lung injury [20], alveolar liquid clearance, respiratory virus infection [21] and virus isolation [22] *Mycoplasma pneumoniae* and ENaC targeting drug discovery [12]. To date, there is no clinically used blocker of ENaC. The hypothesis of exploring an ENaC-blocking method to facilitate restoration of the airway surface liquid (ASL) volume sufficiently to allow normal mucociliary clearance is of interest in the management of lung disease in cystic fibrosis patients.

TNF has dual role in pulmonary edema reabsorption [10]. It has been shown that activation of TNF receptor 1 inhibits the transcription of ENaC subunits [11]. In A549 cells TNF has shown to increase Na^+^ uptake in a catecholamine-independent manner [23]. TNF-induced activation of the ENaC whole-cell current in A549 cells has also been shown [12]. Additionally, δ-ENaC is proved to play an important role in the TNF lectin-like domain derived peptides-induced ENaC activation [13]. Although A549 cells are widely accepted model in ENaC research, the precise role of δ subunit in the ENaC channel conduction has not been studied.

Using new CRISPR knock out technology we have shown for the first time that the transfection of CRISPR knock out plasmid significantly reduced the protein expression of δ-ENaC in both western blot and immunofluorescence assays as compared to control plasmid transfected cells (Figure 1A,B). The δ-ENaC knock down in the A549 cells decreased significantly the whole-cell Na^+^ current in control and ameliorated the TNF-induced activation of ENaC (Figure 2A,B), indicating that δ-ENaC plays a major role in the TNF-induced activation of ENaC in the A549 cells.

Overexpression of WT δ-ENaC back in the KD A549 cells restored the ENaC control current and TNF-induced activation of ENaC (Figure 3A,B). The whole-cell current amplitude after 10 nM TNF treatment in these WT δ-ENaC transfected KD A549 cells was even higher than control A549 cells, indicating that δ-ENaC plays a major role in the TNF-induced activation of ENaC current in A549 cells. This restoration of whole cell ENaC current could be due to the increase in the open probability, single channel amplitude or both.

It has been previously shown that TNF has dual role on the ENaC [10]. It exerts its inhibitory effects on ENaC subunits transcription through TNF receptor 1 [11] or increases the activity of ENaC through its lectin-like domain [24]. We have also shown that TNF lectin-like domain mimicking peptides can activate the ENaC directly [13,16] in overexpression system and A549 cells [12] and in vivo experiments [25]. Additionally, we found that TNF lectin-like domain exerts its effect by docking on the glycosylation sites of ENaC subunits and then through carboxyl terminus of ENaC subunits [16,18]. To analyze if δ-ENaC KD A549 require glycosylation sites and carboxyl terminus of the δ-ENaC, we overexpressed the δ-ENaC (3NQ) and δ-ENaC (D522X). Surprisingly, the N-linked glycosylation sites mutated δ-ENaC was able to restore the TNF-mediated ENaC activation in the ENaC KD A549 cells (Figure 4A,B). The possible interpretation of these results could be that in A549 cells fully glycosylated αβγ-ENaC are also expressed which might have compensated the loss of glycosylation sites of δ-ENaC. Additionally, when δ-ENaC KD A549 cells where transfected with the N-linked glycosylation sites mutated and carboxyl terminus deleted δ-ENaC, TNF-induced activation of whole-cell ENaC current was again restored (Figure 5A,B). A possible interpretation of these results could again be that other αβγ subunits might have compensated the loss of carboxyl terminus domain of δ-ENaC.

Taken together, the results indicate that, in A549 cells, the presence of δ-ENaC is required for the TNF-induced activation of whole-cell ENaC currents. However, glycosylation of the δ-ENaC and deletion of carboxyl terminus were not necessary for the TNF activation of ENaC currents.

## 4. Materials and Methods

### 4.1. Cell Culture

Human lung adenocarcinoma A549 cells (ATCC no. CCL-185) in passaged 80–97, were seeded in Dulbecco’s modified Eagle medium/F12 nutrient mixture Ham plus L-glutamine (DMEM/F-12; Gibco™ by Life Technologies, Vienna, Austria), supplemented with 10% fetal bovine serum (FBS; Gibco™ by Life Technologies, Vienna, Austria) and 1% penicillin-streptomycin (Sigma-Aldrich, Vienna, Austria). Cells were maintained at 37 °C with 5% CO_2_ in a humidified incubator.

### 4.2. Transfection of CRISPR/Cas 9 Plasmids

A549 cells were transfected with CRISPR/Cas9 KO plasmid from Santa Cruz sc-404577 of control plasmid sc-418922 one day after seeding using X-tremeGENE HP DNA transfection reagent (Roche Diagnostics, Mannheim, Germany) according to the manufacturer’s protocol. The expression was highest 48 to 72 h after transfection. cDNAs encoding for WT subunits of hENaC and 3NQ δ-ENaC and δ3NQ-D522X-ENaC are described previously [18]. δD522X-ENaC is missing 37 amino acid long COOH terminus.

### 4.3. Immunofluorescence and Confocal Imaging

For immunostaining and imaging, A549 cells were cultivated on coverslips. The cells were fixed with 4% paraformaldehyde (PFA) for 20 min, permeabilized by 0.02% PBST (PBS with Triton X-100) for 20 min, washed 3X with PBS and blocked with 3% BSA in PBS for 30 min at room temperature. Anti-δ-hENaC antibody (1:500) was applied for 1 h at RT. After washing, cells were incubated with secondary antibody (goat anti-rat Alexa Fluor 647, Thermo Fisher Scientific), Vienna, Austria, for 1 h at RT in dark. Confocal images were acquired with a Zeiss LSM-510 confocal laser scanning microscope. Quantification was performed by ImageJ U.S. National Institutes of Health, Bethesda, Maryland, USA.

### 4.4. Western Blotting

A549 cells were grown in 10 cm dishes in 37 °C, 5% CO_2_ incubator in DMEM medium supplemented with 5% FBS. A549 cells transfected with either CRISPR/Cas 9 control plasmid (control) or CRISPR/Cas 9 δ-ENaC KO plasmid (KD). Then cells were washed twice with 10 mL ice-cold PBS. Following treatment, cells were washed twice with ice-cold PBS before extracting buffer (150 mM sodium acetate, 0.9% NaCl, 0.1% Triton X-100, pH 5.5) was applied. After scraping cells off the dishes, the resulting cell suspension was collected in vials and lysed using ultrasonication (3 × 10 s). Following incubation on ice for 30 min, cell debris was removed by centrifugation at 13,000 RPM/4 °C for 15 min. The supernatant subjected to protein electrophoresis and immunoblotting. The proteins were separated under reducing conditions by SDS-PAGE using 7.5% SDS gel along with prestained protein marker (cat. #12949 from Cell Signaling). Proteins were then transferred onto a nitrocellulose membrane (UltraCruzTM 0.45 mm, Santa Cruz Biotechnology, Dallas, Texas, USA) by semi-dry blotting at 25 V for 30 min. Unspecific binding sites were blocked by incubating the membrane overnight at 4 °C with 3% FBS in PBS supplemented with 0.02% sodium azide. Membrane was then incubated for 90 min with primary antibody anti-δ-hENaC and anti-β-actin from Sigma Aldrich. Membrane was washed 5× with 10 mL PBS containing 0.1% Tween-20 (PBST) and corresponding horseradish peroxidase-conjugated secondary antibodies (Santa Cruz Biotechnology) were applied. After 90-min incubation, membrane was washed 3× with PBST and once with PBS. Enhanced chemiluminescence (ECL) substrate (Amersham ECL Plus Western Blotting Detection Reagent, GE Healthcare, Vienna, Austria) was used for visualization. Following incubation for 2 min, membranes were exposed to X-ray films (Amersham Hyperfilm ECL, GE Healthcare, San Francisco, CA, USA). Exposed films were scanned and quantified using ImageJ U.S. National Institutes of Health, Bethesda, Maryland, USA.

### 4.5. Electrophysiology

Electrophysiological experiments were performed as described in detail by Shabbir et al. [13]. Briefly, effects of TNF on control and knock down δENaC were studied on transfected A549 cells at room temperature (19–22 °C) 24 to 48 h after plating. Currents were recorded with the patch clamp method in the whole-cell mode. The chamber contained 1 mL of the bath solution of the following composition (in mM): 145 NaCl, 2.7 KCl, 1.8 CaCl_2_, 2 MgCl_2_, 5.5 glucose and 10 HEPES, adjusted to pH 7.4 with 1 M NaOH solution. Micropipettes were pulled from thin-walled borosilicate glass capillaries (Harvard Apparatus, Holliston, MA, USA) with a DMZ Zeitz Puller to obtain electrode resistances ranging from 2 to 5 MΩ. The pipette solution contained (in mM): 135 potassium methane sulphonate, 10 KCl, 6 NaCl, 1 Mg_2_ATP, 2 Na_3_ATP, 10 HEPES and 0.5 EGTA, adjusted to pH 7.2 with 1 M KOH solution. Chemicals for pipette and bathing solutions were supplied by Sigma-Aldrich (Vienna, Austria). Electrophysiological measurements were carried out with an Axopatch 200B patch clamp amplifier (Axon Instruments, San Jose, CA, USA). Capacity transients were cancelled and series resistance was compensated. Whole-cell currents were filtered at 5 kHz and sampled at 10 kHz. Data acquisition and storage were processed directly to a PC equipped with pCLAMP 10.2 software (Axon Instruments, San Jose, CA, USA). After GΩ-seal formation, the equilibration period of 5 min was followed by control recordings at a holding potential of 100 mV to −100 mV with 20 mV increments. Then, aliquots of a stock solution, which was prepared with distilled water, were cumulatively added into the bathing solution. The ENaC activation effect was estimated from currents recorded at E_h_ of −100 mV).

### 4.6. Test Compounds

TNF and all other chemicals, reagents and culture media were obtained from Sigma-Aldrich (Vienna, Austria), unless stated otherwise

### 4.7. Statistical Analysis

Data were analyzed with OriginPro 2017 (OriginLab, Northampton, MA, USA) and figures were edited with CorelDRAW X7 (Corel Corporation, Ottawa, ON, Canada). Data are represented as mean ± SEM of at least three independent biological replicates/experiments. Significant differences of two independent values were evaluated by unpaired Student’s *t*-test. Statistical significance * *p* < 0.05, *** *p* < 0.001 was calculated with the Student’s *t*-test.

## 5. Conclusions

We demonstrated in this study that δ-ENaC is necessary for the TNF-induced activation of ENaC in A549 cells.

## Figures and Tables

**Figure 1 ijms-22-01858-f001:**
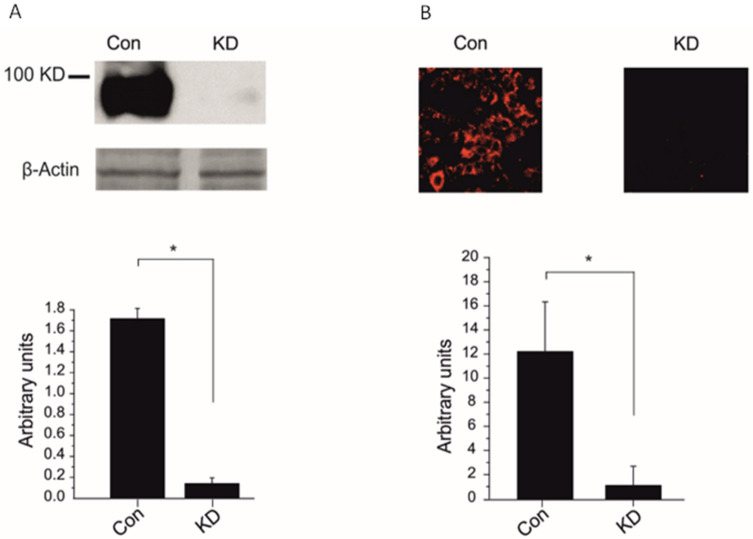
CRISPR/Cas 9 knocked down the δ-ENaC subunit in the A549. (**A**) Upper panel, A549 cell lysate blotted against anti δ-ENaC and β-Actin antibodies. Representative western blot out of three independent biological replicates is shown. A549 cells transfected with either CRISPR/Cas 9 control plasmid (control) or CRISPR/Cas 9 δ-ENaC KO plasmid (KD). The lower panel; mean values of quantitation of three western blots is shown. (**B**) Upper panel, representative confocal microscope image of anti δ-ENaC antibody, is shown. A549 cells transfected with either CRISPR/Cas 9 control plasmid (control) or CRISPR/Cas 9 δ-ENaC KO plasmid (KD). The lower panel; mean values of quantitation of 7 immunofluorescence images are shown. Significant differences are indicated, * *p* < 0.5, unpaired Student’s *t*-test, *n* = 5–7.

**Figure 2 ijms-22-01858-f002:**
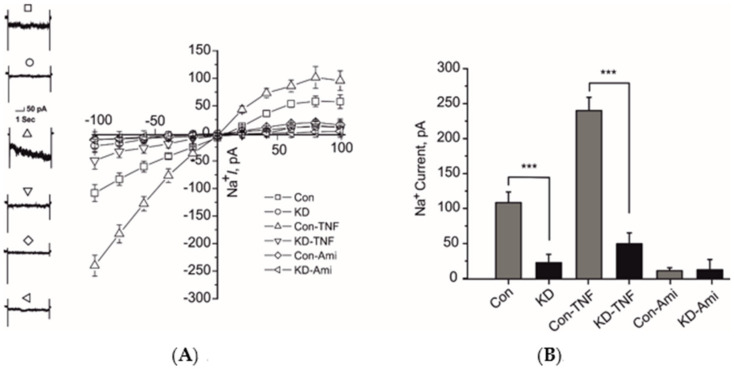
Knock down of δ-ENaC in A549 cells attenuated the tumor necrosis factor (TNF)-induced activation of ENaC. (**A**) Left panel, representative traces of whole-cell (10 µM) amiloride-sensitive ENaC currents at 100 mV of indicated experimental settings are shown. Right panel, IV plots of whole-cell currents of indicated experimental settings are shown. (**B**) Mean values of whole-cell inward sodium currents at −100 mV of indicated experimental settings are shown. Significant differences are indicated, *** *p* < 0.001, unpaired Student’s *t*-test, *n* = 12–15.

**Figure 3 ijms-22-01858-f003:**
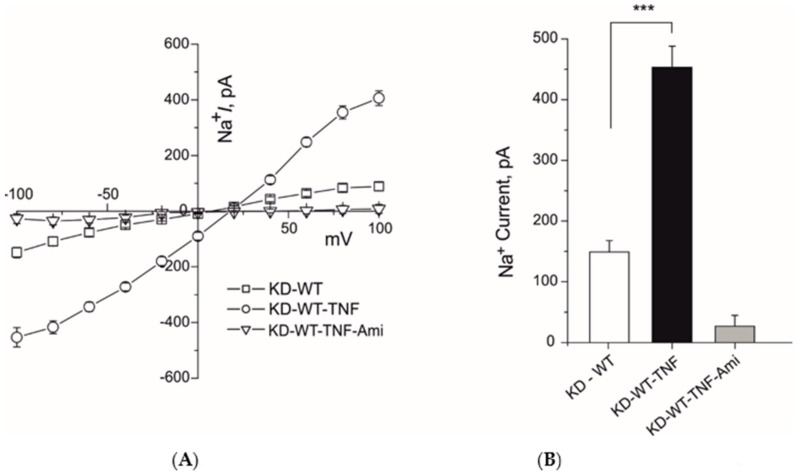
Overexpression of δ wild-type (WT) restored the TNF mediated activation of ENaC in δ-ENaC KD A549 cells. (**A**) left panel, IV plots of whole-cell (10 µM) amiloride-sensitive ENaC currents of indicated experimental settings are shown. (**B**) Mean values of whole-cell inward sodium currents at −100 mV of indicated experimental settings are shown. Significant differences are indicated, *** *p* < 0.001, unpaired Student’s *t*-test, *n* = 12–15.

**Figure 4 ijms-22-01858-f004:**
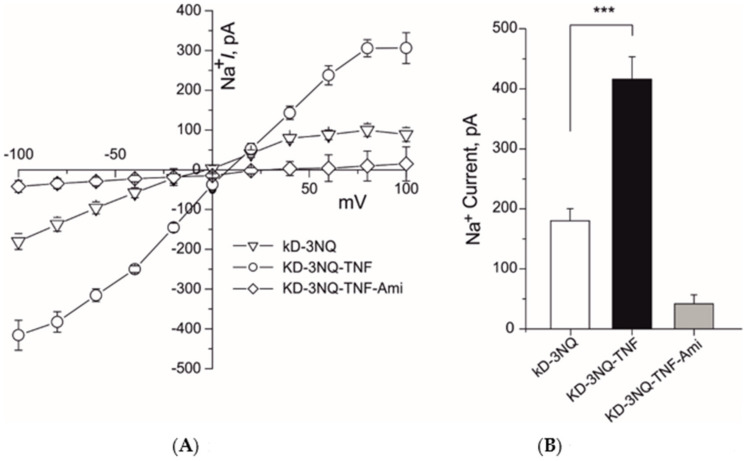
Overexpression of N-linked glycosylation sites mutated δ-ENaC (3NQ) restored the TNF mediated activation of ENaC in δ-ENaC KD A549 cells. (**A**) IV plots of whole-cell (10 µM) amiloride-sensitive ENaC currents of indicated experimental settings are shown. (**B**) Mean values of whole-cell inward sodium currents at −100 mV of indicated experimental settings are shown. Significant differences are indicated, *** *p* < 0.001, unpaired Student’s *t*-test, *n* = 12–15.

**Figure 5 ijms-22-01858-f005:**
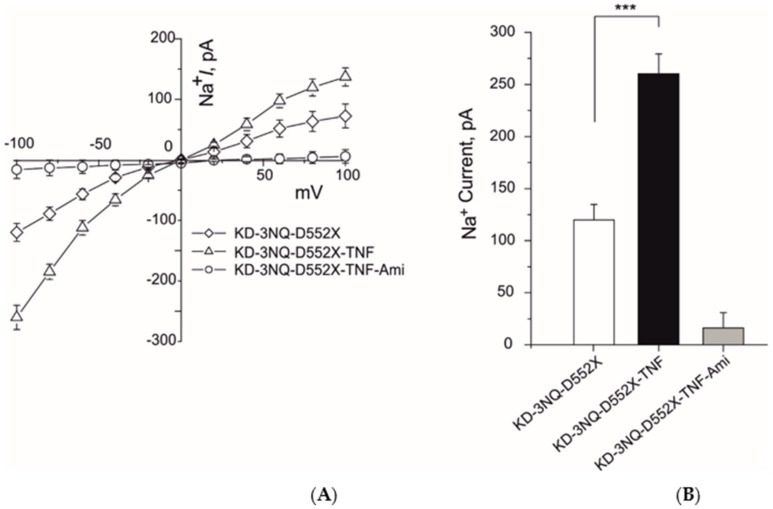
Overexpression of N-linked glycosylation sites mutated and carboxyl terminal dele δ-ENaC (3NQ-D522X) restored the TNF mediated activation of ENaC in δ-ENaC KD A549 cells. (**A**) IV plots of whole-cell (10 µM) amiloride-sensitive ENaC currents of indicated experimental settings are shown. (**B**) Mean values of whole-cell inward sodium currents at −100 mV of indicated experimental settings are shown. Significant differences are indicated, *** *p* < 0.001, unpaired Student’s *t*-test, *n* = 9–11.

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
