# Peer review of "CRISPR/Cas9 Mediated Knock Down of δ-ENaC Blunted the TNF-Induced Activation of ENaC in A549 Cells"

_ijms, 2021, doi:10.3390/ijms22041858_

Round 1

Reviewer 1 Report

In this manuscript, Waheed et al., have shown that δ-ENaC is expressed significantly in a human lung adenocarcinoma A549 cell line. CRISPR/cas9 mediated knockdown of this subunit markedly blunted the TNF-induced activation of ENaC currents, indicating a primary role for δ-ENaC in TNF induced activation in A549 cells. This is a nice, well-written and focused manuscript. The experiments are well-designed and the data supports the conclusions. I have few minor concerns. For a general reader:

  1. It will be helpful to include little background related to the tissues/organs in which this ENaC and specifically δ-subunit is expressed, and role of ENaC channels.
  2. Authors should also mention in the background or discussion whether the increase in current due to δ-subunit is related to increase in single channel current amplitude, change in open channel probability or any other unique property of δ-subunit.
  3. D522X mutant needs to be described in more detail. It is not clear how long is the C-termini and how big is the truncation.
  4. Fig. 2A, representative current traces are too small to see clearly.

Author Response

Dear Reviwer 1

Thank you so much for reviewing our manuscript and giving us your valuable suggestions. We addressed all of your minor concerns in the revised manuscript file. Please find the answer to your comments below.

  1. It will be helpful to include little background related to the tissues/organs in which this ENaC and specifically δ-subunit is expressed, and role of ENaC channels.

Answer: It is a good pick. We provided more details about the expression tissue distribution and function of δ-subunit of ENaC with new citations. The new text can be seen in the revised manuscript file from line 29-49.

2. Authors should also mention in the background or discussion whether the increase in current due to δ-subunit is related to increase in single channel current amplitude, change in open channel probability or any other unique property of δ-subunit.

Answer: This is a very good point. thank you so much for pointing it out. We mentioned in the discussion that the restoration of whole-cell ENaC current could be due to the increase in the open probability, single-channel amplitude, or both. Please see lines 214-215.

3. D522X mutant needs to be described in more detail. It is not clear how long is the C-termini and how big is the truncation.

Answer: We describe the D552X in more detail that this subunit was lacking a 37 amino acid COOH terminus domain, Please see line 252.

4. Fig. 2A, representative current traces are too small to see clearly.

Answer: We enlarged the traces in Fig. 2A, new Fig 2A can be seen in the revised manuscript.

We again thank you so much for these valuable suggestions. We feel that these changes have improved the quality of our work. the authors thank you so much for that.

With kind regards,

Dr. Waheed Shabbir

Reviewer 2 Report

The authors explored the role of δ-ENaC in TNF-induced activation of ENaC in A549 cells. The experimental designs are adequate and the conclusions are sound. The writing is easy to follow, except for some small details to edit. I thus recommend publication with minor revision.

(1) Line 13: advise to be changed to "A549 cells are widely used as a model for ENaC research".

(2) Consider more introduction about A549 cells.

(3) CRISPR/Cas9 approach should be "knock out" instead of "knock down". The question is whether the generated cells are homozygous or heterozygous. Also it is better to show the Cas9 cutting site of δ-ENaC gene for knockout.

(4) Figures whenever there is n number, should be clearer on independent sample or independent experiments. Whenever P value is showed, need to specify the statistical method.

Author Response

Dear Reviewer2

Thank you so much for reviewing our Manuscript and giving us your valuable suggestions. We addressed all of your minor concerns as stated below.

  • Line 13: advice to be changed to "A549 cells are widely used as a model for ENaC research.

Answer: We accepted the advice and the sentence was changed, please see line 47.

(2) Consider more introduction about A549 cells.

Answer: We added more introduction about A549 cells, please see lines 37 to 47.

3) CRISPR/Cas9 approach should be "knock out" instead of "knock down". The question is whether the generated cells are homozygous or heterozygous.

Answer: This is a very good point. However, this study was not amid to carry on with the knocked down (knocked out) cells therefore we just mentioned knocked down. We did not perform any further selection in this case.

Also it is better to show the Cas9 cutting site of δ-ENaC gene for knockout.

Answer: We checked the datasheets of Santa Cruz for these details, but it seems that they don’t want to disclose this information. All we know that this CRISPR-Cas9 knockout plasmid was consists of three different guide RNAs to achieve the maximum efficiency, which we were able to achieve.

(4) Figures whenever there is n number, should be clearer on independent sample or independent experiments. Whenever P-value is shown, need to specify the statistical method.

This is again a very good point. thank you so much for pointing it out. We mentioned the statistical method with all the P values. Please see lines 153, 161, 168, 174, 180.

We again thank you so much for suggesting us these minor concerns. We feel that these changes have improved the quality of the manuscript.

With kind regards,

Dr. Waheed Shabbir